# NGS read classification using AI

**Benjamin Voigt**[ID]○, **Oliver Fischer**[ID]○, **Christian Krumnow, Christian Herta**‡,
**Piotr Wojciech Dabrowski**[ID]‡*

Center for Bio-Medical image and Information processing (CBMI), HTW University of Applied Sciences,
Berlin, Germany

○ These authors contributed equally to this work.
‡ CH and PWD also contributed equally to this work.
* piotr.dabrowski@htw-berlin.de

**Data Availability Statement:** All data files are available from zenodo (accessions 4306240, 4306248, 4307779, 4306420, 4306499). The source code is available from github: https://github.com/CBMI-HTW/TaxonomicClassification-NGS-NN.

## Abstract

Clinical metagenomics is a powerful diagnostic tool, as it offers an open view into all DNA in a patient's sample. This allows the detection of pathogens that would slip through the cracks of classical specific assays. However, due to this unspecific nature of metagenomic sequencing, a huge amount of unspecific data is generated during the sequencing itself and the diagnosis only takes place at the data analysis stage where relevant sequences are filtered out. Typically, this is done by comparison to reference databases. While this approach has been optimized over the past years and works well to detect pathogens that are represented in the used databases, a common challenge in analysing a metagenomic patient sample arises when no pathogen sequences are found: How to determine whether truly no evidence of a pathogen is present in the data or whether the pathogen's genome is simply absent from the database and the sequences in the dataset could thus not be classified? Here, we present a novel approach to this problem of detecting novel pathogens in metagenomic datasets by classifying the (segments of) proteins encoded by the sequences in the datasets. We train a neural network on the sequences of coding sequences, labeled by taxonomic domain, and use this neural network to predict the taxonomic classification of sequences that can not be classified by comparison to a reference database, thus facilitating the detection of potential novel pathogens.

## Introduction

Over the past one and a half decades, Next Generation Sequencing (NGS) has revolutionized genomics and adjacent fields of research. The ability to sequence massive amounts of DNA at ever-decreasing costs per base has led to an explosion of the genetic information available for researchers. For instance, since the introduction of the Roche 454, the first commercially successful NGS machine [1], in 2005, the number of bases in GenBank has grown from about $10^{10}$ to almost $10^{12}$, at a staggering average rate of $5 \times 10^{10}$ bases per month—the same number of bases every two months that it had previously taken 22 years to accumulate [2]. And this is just the analyzed tip of the iceberg: The Sequence Read Archive (SRA) currently holds over $4 \times 10^{16}$ bases of raw NGS data [3].

**Funding:** BV and OF are funded by the Federal Ministry of Education and Research of Germany (BMBF, https://www.bmbf.de/) in the project deep. Health (project number 13FH770IX6). The funders had no role in study design, data collection and analysis, decision to publish, or preparation of the manuscript.

**Competing interests:** The authors have declared that no competing interests exist.

This massive amount of available data presents diverse challenges when it comes to data analysis. One common application of NGS is metagenomic sequencing, where all genetic material in a complex sample, such as a patient's body liquid (clinical metagenomics) or a piece of arctic ice (environmental metagenomics), is sequenced. While in targeted approaches, such as sequencing a cultured bacterium, the composition of the sample is known a priori and that knowledge can be used to inform the analysis, in metagenomic sequencing the primary data analysis task is determining that composition. The common approach to this challenge is using known reference sequences. Very broadly speaking, for each read from an NGS sample the similarity to all sequences within a reference database is determined and the read is classified as belonging to a taxon based on this comparison. While this approach allows highly successful detection of organisms with already sequenced relatives, as evidenced by the results of studies such as CAMI where all entries used variations on the above-mentioned approach [4], it does not allow the detection of entirely novel organisms—especially if those are only represented at low levels in the sample. Data from such organisms disappears within the thousands to millions of "unclassified" reads that remain as a byproduct of any metagenomic NGS dataset analysis—hidden among technical artifacts and reads from known organisms that could not be clearly assigned to a taxon.

Although such "unclassified" reads are usually discarded within the analysis workflow, the hints towards novel bacteria and viruses contained therein would be a valuable resource if they could be identified and isolated for further, more detailed analysis. This would facilitate the detection and characterization of the huge number of organisms that have not yet been sequenced—for instance, [5] estimate that there are around $10^{14}$ bacterial taxa, of which only around $10^6$ have been described, and [6] estimate that there are over $3 \times 10^5$ undetected mammalian viruses.

It has previously been shown that machine learning can be a valuable tool in overcoming this challenge: [7] have successfully used Random Forests to predict the presence of sequences from human pathogenic organisms in metagenomic datasets. While several other approaches have used machine learning to optimize parameters of existing tools [8, 9], to our knowledge the work by [7] represents the only attempt to detect microbial sequences using machine learning without relying on reference sequences. Here, we aim to extend the understanding of such approaches' usefulness by applying transformer neural networks to the classification of NGS reads as mammalian, bacterial or viral in origin.

The application of neural network models is profoundly successful in the field of natural language processing [10, 11]. In particular, models based on the Transformer architecture [12] have led to significant breakthroughs in developing so-called language models that have shown state-of-the-art performance on a variety of natural language processing tasks [13–15]. One tremendous advantage is that such models can be trained using a self-supervised approach, i.e., there is no requirement for labeled data like in a supervised learning regime.

In recent work, language models from various transformer networks, primarily multi-layer bidirectional Transformer [14], have been trained on datasets containing a large number of unlabeled protein sequences, e.g., UniProt [16]. [17] developed the protein generation language model (ProtGen) that creates proteins that exhibit near-native structure energies. [18] investigated the learned representations of a protein language model. Their findings show that high-capacity networks reflect biological structure at multiple levels, including amino acids, proteins, and evolutionary homology. [19] compared the performance of the embeddings generated by several network architectures [14, 20–22] on multiple supervised learning tasks, e.g., classification into membrane and non-membrane proteins.

In this paper we build upon the aforementioned work on the application of transformer networks in protein classification to demonstrate their applicability to taxonomic classification

of NGS reads. Since the overarching goal is the detection of entirely novel organisms from metagenomic datasets, in this initial work we focus on a classification on the domain level, specifically into mammalian (i.e. host in the case of clinical metagenomic datasets), viral and bacterial reads. This will allow extraction and specific examination of reads representing hitherto undescribed viruses and bacteria from reads that remain in the "unclassified" bin after traditional metagenomic data analysis and taxonomic classification has been performed with tools such as Kraken [23], RIEMS [24], PathoScope [25], PAIPline [26] or MetaMeta [27]. Since we are building upon large existing models that have been trained on protein sequences, we limit this proof of concept to the classification of reads that lie within a coding sequence (CDS). While, in order to be able to correctly perform classification independent of a read's offset within the CDS, we also automatically determine which of the six possible frame the read is in. This pre-requisite step in itself is a novel application of machine learning to ORF detection, as current tools either (i) rely purely on presence/absence of start/stop codons without further interpretation of the sequence (such as getorf [28] or OrfM [29]) or return all candidate sequences for each read without clearly resolving potentially contradictory hypotheses (such as FragGeneScan [30], CNN-MGP [31] or geneRFinder [32]). However, since this is a proof-of-concept work, we do not—in contrast to these existing tools—examine reads that are outside of CDSs in this paper.

## Methods and implementation

We developed a proof-of-concept for the classification of NGS-reads into the taxonomic domains viruses, bacteria, and mammals. The classification is done by concatenating multiple data processing and sub-classification steps.

At first, the frame of a read within its CDS must be recognized to translate the DNA sequence fragments into amino acid sequences correctly. This is a non-trivial step because there are typically no start or stop codons in the fragments. We developed a classifier based on a language model to detect the correct frame of a read. Using this information, the read sequences can be translated into amino acid sequences. In a final step, the amino acid sequences are classified into taxonomic domains by another language model.

In this section, we describe this proof-of-concept pipeline in more detail. Then, we provide information about the design and training process of the individual classifiers which are used in different steps of the pipeline. Finally, we describe how training and test data sets were generated.

### Classification pipeline

The pipeline was implemented as an python script (see https://github.com/CBMI-HTW/TaxonomicClassification-NGS-NN). A general overview is shown in Fig 1.

As input the pipeline gets a file in the *FASTA* format with NGS reads. We expect the nucleotide sequences $\mathbf{s}$ to be of 300 base pairs each, i.e., $\mathbf{s} = (s_1, \ldots, s_{300})$. This is not a strict requirement as shown by our experiments with different amino acid sequence length (S1 Appendix; S1 Fig), but since the classifiers were optimized on that length it will lead to the best results. While for the prototype, we assume that all reads lie within CDSs, we plan to add an automatic selection of such reads from raw NGS data in future work. Each nucleotide sequence is translated into a protein sequence $\mathbf{x} = (x_1, \ldots, x_{100})$ using biopython [33]. Often, this is not the correct translation because most reads are off-frame. Such wrong translations are detected in the next step and are then re-translated. However, if there are any stop codons in the initial translation, a full six-frame translation of the read is performed and, if a translation $\hat{\mathbf{x}}$ without stop codons is found, this $\hat{\mathbf{x}}$ is used instead.

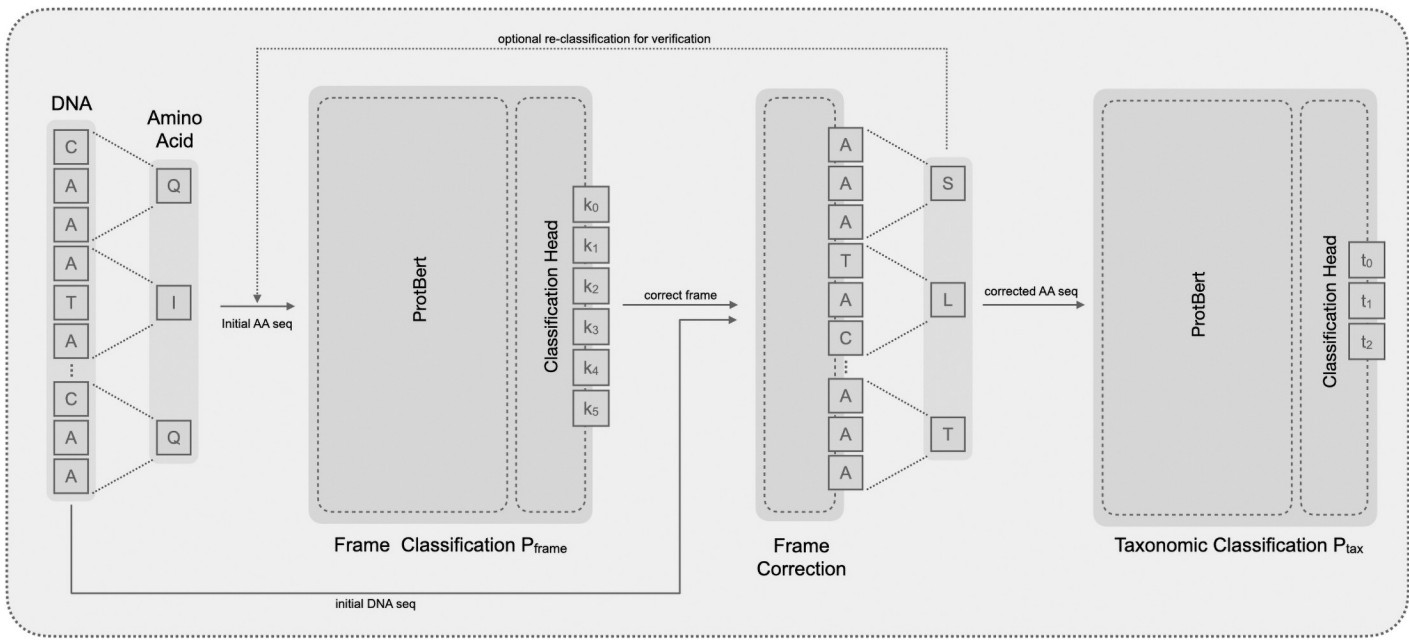

**Fig 1. Overview of the complete neural network classification pipeline.** The pipeline consists of four major blocks: (1) preprocessing NGS reads, (2) frame classification of NGS reads, (3) frame correction and translation of NGS reads, (4) taxonomic classification of amino acid sequence. The dotted arrow line shows an optional loop of the frame classification used for checking the frame correction block, as shown in Fig 2.

The resulting amino acid sequence $\mathbf{x}$ from the initial translation is fed into the frame classification model that returns a probability distribution over the classes $p_{frame}(k \mid \mathbf{x})$. There are six possible classes $k \in \{0, \ldots, 5\}$ which are predicted by the model for each sequence: on-frame ($k = 0$), offset by one base ($k = 1$), offset by two bases ($k = 2$), reverse-complementary ($k = 3$), reverse-complementary and offset by one base ($k = 4$) and reverse complementary and offset by two bases ($k = 5$). Based on the classification result $\hat{k}$, with $\hat{k} = \arg\max_k p_{frame}(k \mid \mathbf{x})$, the transformations (shifting, reverse-complementing) necessary to make each amino acid sequence on-frame are performed on the original DNA sequences $s$ before they are again translated into amino acid sequences $\dot{\mathbf{x}}$ for further processing. Finally, each on-frame amino acid sequence $\dot{\mathbf{x}}$ is classified into one of the taxonomic domains $t$ used in this prototype: *virus*, *bacteria*, or *human/mammalian* using the second classification model $\hat{t} = \arg\max_t p_{tax}(t \mid \dot{\mathbf{x}})$.

The output of the pipeline is the input *FASTA* file with a modified identifier line, i.e., information about the frame classification $\hat{k}$ and species classification $\hat{t}$ is appended.

## Model description, training, and fine-tuning

The classifiers are based on pre-trained (protein) language models. A language model is an assignment of a probability distribution to a sequence of tokens for a language. In our context, the tokens are amino acid symbols. Such language models are trained (self-supervised) on large corpora of sequences/sentences of a language, e.g. on protein data bases. In the training process the language model must continue a sequence (autoregressive) or should predict missing tokens which are masked in the input [14], i.e. it must assign high probabilities to the corresponding tokens of the training data. Therefore, no labeled data is needed.

A neural network language model can be used after training as a *feature extractor*, i.e., the token sequence is transformed by the model into a sequence of feature vectors. The sequence

of features can be used for other tasks, e.g., for the classification of a complete sequence. Herefore, the sequence of features is pooled into one feature vector with a fixed size. With additional labeled data a classification model can be trained on pairs of feature vectors and labels. This is an example of *transfer learning*: The knowledge learned by modeling the language is used for another task, e.g., a classification. The parameters of the feature extractor can also be modified for the specific task. Typically, a classification head replaces the last layer(s) of the pre-trained language model neural network such that the new neural network can be trained end-to-end on the classification data (sequence-label pairs). I.e., the parameters of the neural network are *fine-tuned* to improve the classification performance.

We used a pre-trained language model called ProtBert [19], a multi-layer bidirectional transformer neural network [14] with 30 layers and 420 million parameters. The language model was trained on the Uniref100 dataset [16]. As an important detail we note that the dataset contains protein sequences in their correct reading frame. Such sequences don't contain any stop codon which we take into account in our pipeline as discussed above in Section "Classification Pipeline" and in our data generation as described in Section "Data generation".

The language model maps an amino acid sequence $\mathbf{x} = (x_1, \ldots, x_n)$ into a sequence of feature vectors $(\mathbf{h}'_1, \ldots, \mathbf{h}'_n)$. In case of the pre-trained language model ProtBert each $\mathbf{h}'$ consists of 1024 elements. We reduce these sequence of feature vectors into a single feature vector $\mathbf{h}$ by using different pooling strategies. We explored mean, max, and dot product self-attention functions for pooling. For classification, we fed the feature vector $\mathbf{h}$ into a two-layer dense network (classification head), projecting this representation into unnormalized log probabilities $\mathbf{z} = (z_1, \ldots, z_c)$ with $c$ being the number of classes of the task. The probabilities are computed from $\mathbf{z}$ by a softmax operation. Note that the language models are trained on complete protein sequences. In contrast, the classification is done on (protein) fragments.

Both classification models $p_{frame}$ and $p_{tax}$ were trained with two different approaches: (pure) feature extraction and fine-tuning. In the first variant, the feature vectors generated by the transformer were fixed, i.e., the parameters that were trained by the language model objective are frozen. Only, the parameters of the classification head are adapted during training. This approach has the advantage of significantly reduced training time. Since only the small dense network parameters have to be updated, the batch size can be increased. However, updating all weights during the training process lets the pre-trained models adapt to the specific task. We explored this alternative using smaller batch sizes and different learning rates for the dense networks and the pre-trained language model. Regardless of the training type, we used the LAMB optimizer [34] to update the model parameters. We optimized model hyperparameters using the the ASHA algorithm [35]. We summarize the final hyperparameter settings of our experiments in Table 1.

**Table 1. Hyperparameter settings used for the training process of $P_{frame}$ and $P_{tax}$.**

|  | $P_{frame}$ | $P_{tax}$ |
|---|---|---|
| Epochs | 2 | 10 |
| Batch size | 128(8) | 64(4) |
| LM Learning Rate | $1e^{-5}$ | $25e^{-5}$ |
| CN Learning Rate | 0.0025 | 0.0005 |
| CN Feature Number | 512 | 256 |
| CN Dropout Rate | 0.0 | 0.25 |

For brevity in the table *LM* is short for language model and *CN* referring to the classification network.

The model calculation was done using a small cluster consisting of 8 *Nvidia V100* GPUs. We realized distributed training through data parallelism, i.e., distribute the same model with different batches across the nodes. In Table 1 we report the global batch size with the number of nodes in brackets if calculated distributed, e.g., 256(4), meaning a batch size of 64 per GPU.

## Data generation

To train the models as described above with the most reliable data available, we used amino acid sequences from Uniprot [16] for the taxonomic classification model and RefSeq [36] reference sequences for the frame classification model. We then tested the applicability of the trained models and the whole pipeline by classifying reads from two metagenomic sequencing projects available within the SRA [37], after selecting only the reads matching the criteria for this pipeline (i.e. those that lie within a viral, bacterial or mammalian CDS). The steps for generating the input sequences **x** for the classifiers from the initial sequences **s** of the three raw data sets are described in detail in the following sections.

**Training data for taxonomic classification.** For training of the taxonomic classification, we downloaded the 2020–04 release of the fully manually annotated and curated UniProtKB/ Swiss-Prot database for bacteria, viruses and human as representative for mammalian sequences [16, 38]. For each amino acid sequence $\mathbf{x} = (x_1, \ldots, x_{N(\mathbf{s})})$ we created with a sliding window all possible patches $(x_l, \ldots, x_{l+99})$ for all $l \leq N(\mathbf{s}) - 100$ of length 100. $N(\mathbf{x})$ denotes the varying length of the initial sequence **x** and sequences with $N(\mathbf{x}) < 100$ were discarded.

The initial data is strongly unbalanced with respect to its biological origin. In order to split the data in training, validation and test sets we iteratively draw without replacement from all sequences **x** and fill all patches generated from **x** successively either in the test, validation or training. Further we balanced the data by considering all viral sequences and down sampling sequences with bacterial and human origin until all data sets contain the same number of patches for all three classes with an approximate ratio of (10% test, 10% validation, training 80%) of the total sizes.

The final data sets contain approximately $1.8 \times 10^7$ patches and are deposited at zenodo [39].

**Training data for frame classification.** For training of the frame classification, randomly selected viral and bacterial genomes and the human (GRCh38.p13) reference genome were downloaded from GenBank [40]. From these genomes, all annotated CDS DNA sequences **s** were extracted. Similar to the amino acid data, for each nucleic acid sequence $\mathbf{s} = (s_1, \ldots, s_{N(\mathbf{s})})$ using a sliding window all possible patches $(s_l, \ldots, s_{l+299})$ of length 300 as well as their reverse complemented versions $(\overline{s_{l+299}}, \ldots, \overline{s_l})$ for all $l \leq N(\mathbf{s}) - 300$ were created and translated to amino acids using biopython [33]. By this, we create the on-frame sequence, as well as all possible off-frame configurations.

In order to avoid the model from relying on the presence of a stop codon for the classification of off-frame sequences, all sequences whose translation contained a stop codon were discarded. We split the data into three sets with approximate ratios of (10%, 10%, 80%) of the total sizes by placing all patches generated from one initial sequence **s** into one of the three sets.

Due to the removal of sequences containing stop codons which are only present in off-frame sequences, on-frame sequences were heavily over-represented in these data sets. We balanced the data sets by discarding sequences in over-represented frames until all frames were present at the same ratio—and the final three data sets have the exact size ratios (10% test, 10% validation, 80% training). The resulting data sets contain a total of $1.2 \times 10^7$ patches and are deposited at zenodo [41].

**Application data.** To test the applicability of the trained models to real data, we downloaded the raw NGS reads from two metagenomic SRA runs using a read length of 300: A human skin metagenomic study (SRR7188139) and a swine feces metagenome (ERR3013343). Since the proof-of-concept pipeline only classifies reads that lie within a CDS, eligible reads were extracted by mapping. To that end, all RefSeq viral and bacterial genomes, the human reference genome (GRCh38.p13) and the sus scrofa reference genome (GCF_000003025.6) were downloaded, annotated CDS sequences were extracted and raw reads were mapped to these using bowtie2 [42] with the –end-to-end parameter. Reads mapping to either only viral, bacterial or mammalian ORFs were selected for the application test.

## Benchmarking of frame classification

The first step of the pipeline—determining the frame for translating a read's sequence—is a task that is also tackled by other existing tools. It is therefore not immediately obvious whether the best performance can be achieved by using frame classification using ProtBert, as shown in Fig 1, or by using one of these existing tools. In order to answer this question, we have compared the ability of our classifier to predict the correct frame of a read to that of other tools.

We have found six tools that tackle the task of determining the correct translation frame directly from short NGS reads: MGC [43], metaGun [44], metaGene [45], Orphelia [46], fragGeneScan [30] and CNN-MGP [31]. The last one of these, CNN-MGP, also uses a neural network to perform the classification. Unfortunately, out of these six tools, only two were suitable for our comparison. Orphelia requires a java binary that was built against gcc version 3.4, which has been superseded by version 4.0 in late 2006. Setting up a system with such old package versions was outside of the scope of this work. The websites referenced in the publications for metaGun and metaGene are offline, and MGC does not even mention any download website or include the binaries in the supplementary materials. We were not able to find any other resources such as mirrors or git repositories from which source code or binaries can be downloaded, making it impossible to run any of these tools. Accordingly, we only included fragGeneScan and CNN-MGP in our benchmarks.

In order to make the benchmark reproducible, we have implemented it as a nextflow [47] pipeline (see https://gitlab.com/dabrowskiw/cdsfinderbenchmark). For the evaluation, we have used the above-mentioned test dataset [41]. Since the documentation of CNN-MGP output is not entirely clear on how the reported reverse reading frames are encoded, we have manually tested all possible interpretations and chosen the one yielding the best results for CNN-MGP. We have also excluded reads for which CNN-MGP or fragGeneScan reported no reading frame from the calculation, since including these would have given our approach an unfair advantage—while we know that in this proof-of-concept work we have only included reads from within CDS and we thus predict a frame for each read, CNN-MGP and fragGeneScan can be applied to real data also including reads from non-coding regions and thus need to be able to predict that a read contains no valid frame.

## Results

In this section, we report the results of the trained model for two different settings. First, we test the taxonomic and frame classification models separately on the test data of the corresponding training setups. Then we use the full pipeline on real data from metagenomic sequencing studies.

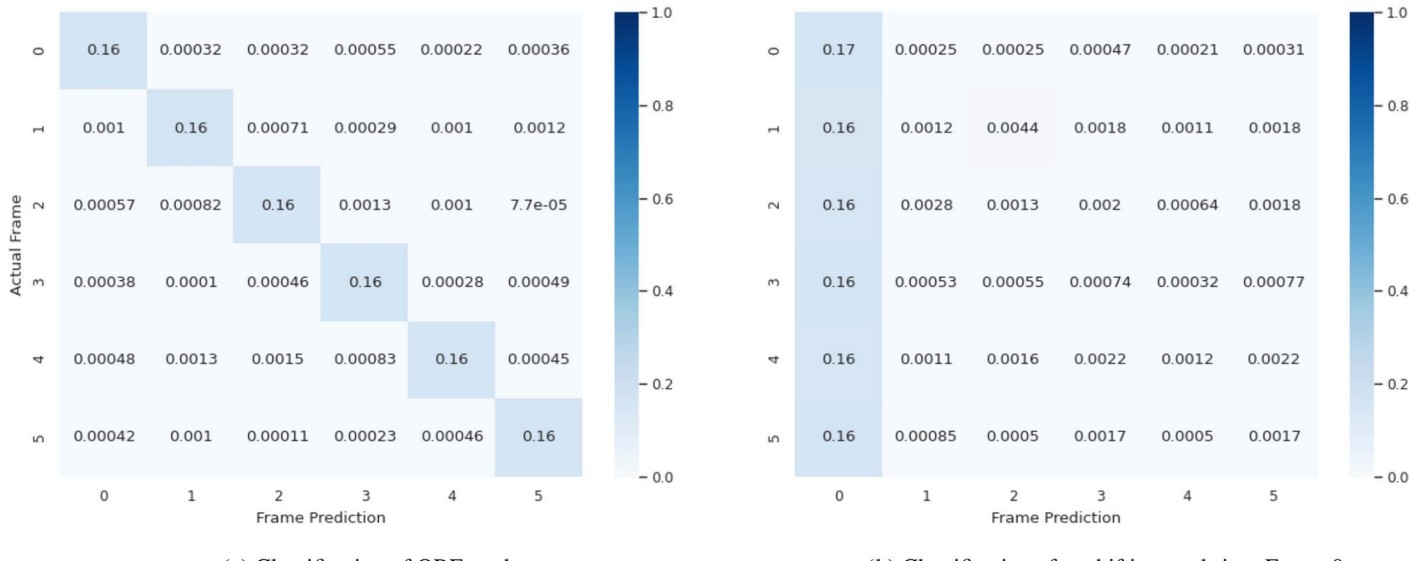

(a) Classification of ORF reads                           (b) Classification after shifting reads into Frame 0

**Fig 2. Prediction results of the frame classification model on the test dataset.** Predictions of the most probable class $\hat{k}$ on the frame test data [41] are shown as an error matrix. The classes are as follows: on-frame (0), offset by one base (1), offset by two bases (2), reverse-complementary (3), reverse-complementary and offset by one base (4), reverse complementary and offset by two bases (5). (A) represents the initial classification results of the reads. (B) Re-evaluation of the reads after applying the frame correction to validate that the reads were correctly shifted to be on-frame, i.e., $k = 0$.

## Evaluation of both models

We evaluated the final classification models using a 10% split of generated data. As metrics, we report the ROC curve and error matrix as a heatmap for both tasks.

On the test dataset, the frame classification $p_{frame}$ achieved an overall accuracy of 0.98 (S2 Fig in S1 Appendix). Since the datasets' classes are balanced, we expected a strong diagonal in the classification task's error matrix. This expectation has been confirmed; see Fig 2A. After applying the frame correction, we used the classification model to verify that the reads were correctly shifted into frame zero ($k = 0$). We observed that almost all sequences had been moved accordingly (Fig 2B).

We measured an accuracy of 0.91 on the corresponding test data for the taxonomic classification model $p_{tax}$, as shown in the error matrix (Fig 3A). We calculated the inner class accuracies to inspect that result in more detail. We observe that for reads predicted as bacterial, indeed 94% were correctly classified. In contrast, for sequences classified as viral, only 88% were actually of viral origin and 8% were mammalian. We observed a similar behavior in the reads classified as mammalian (92% mammalian and 6% were viral). This indicates that the classifier has the most problems in differentiating between these two classes. This observation is also reflected in Fig 3B, the classifier's ROC curve, where class 0 (viral) and 2 (human) are slightly worse compared to class 1 (bacterial). This is likely due to the presence of retroviral sequences in the human genome.

## Exemplary analyses

The exemplary analysis of data from real metagenomic sequencing studies presented a more challenging classification task, most likely due to more noisy data. Firstly, training and testing was performed using error-free sequences derived from curated references, while real NGS reads contain sequencing errors. Secondly, since filtering of reads belonging to a CDS was

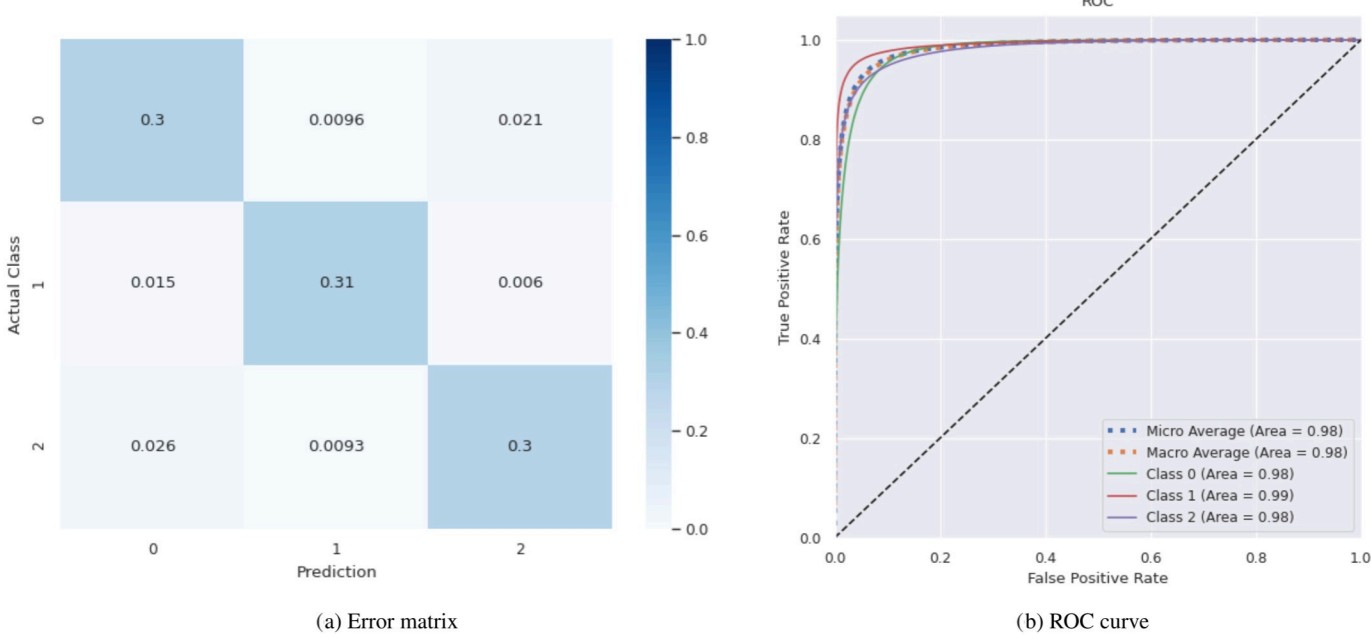

(a) Error matrix                                (b) ROC curve

**Fig 3. Prediction results of the taxonomic classification model on the test dataset.** Error matrix (A) and ROC curve (B) on the taxonomic test dataset [39] are shown. The classes are as follows: 0—viral, 1—bacterial, 2—mammalian.

performed by mapping to all CDS sequences annotated in RefSeq, all automatic annotations that mistakenly classified a non-coding open reading frame as CDS cause reads that do not encode a real protein sequence to remain in the dataset. Still, a taxonomic domain classification precision of 0.87 could be achieved on the swine feces metagenomic dataset, or 0.90 if all reads that had a stop codon in every possible frame were discarded (since this should not happen in a correct read within a CDS and can thus be seen as indicative of an error). These results are visualized in Fig 4A.

Using SKESA [48] with default parameters on the reads classified as viral it was possible to de novo assemble five out of six CDSs of the porcine epidemic diarrhea virus present in the sample, each in a single config—the only CDS that could not be be assembled was that encoding the envelope protein, since it is shorter than the read length of 300 bases used in the analysis and accordingly reads containing this CDS were discarded during dataset generation.

However, the classification of reads from the human skin metagenome only showed a precision of 0.62, due to a large number of bacterial reads being wrongly classified as viral. The resulting ROC curve is shown in Fig 4B. This disparate performance on different datasets warrants closer examination in the future. One possible explanation could be the presence of unannotated bacteriophage sequences on the bacterial reference genomes used for the mapping-based classification of the reads, which would lead to the observed discrepancy between the neural network's and the mapper's assessment of whether a sequence is bacterial or viral in origin.

## Benchmarking of frame classification

In the classification of the frame in a read, our approach using ProtBert (98.18% correctly classified frames) significantly outperformed both CNN-MGP (33.62% correctly classified frames) and fragGeneScan (58.65% correctly classified frames). However, it is important to note that

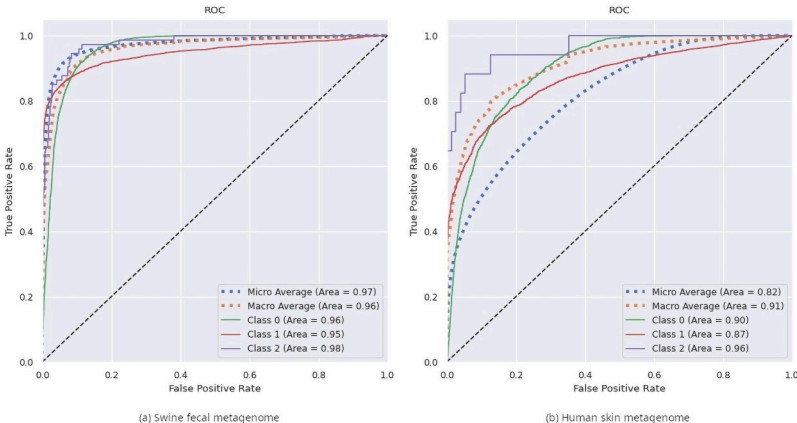

**Fig 4. Prediction results of the complete classification pipeline on data from real metagenomic sequencing studies.** ROC curves for taxonomic classification in swine feces metagenome (A) and human skin metagenome (B) datasets. The classes are as follows: 0—viral, 1—bacterial, 2—mammalian.

due to the limitations described in the methods section, these results are only meaningful in the context of this specific proof of concept work. Especially given the current limitation to recognizing the frame of reads that wholly lie within CDS, this step of the pipeline does not represent a production-ready alternative method of determining the correct frame of NGS reads in general.

## Discussion

With this work, we have shown that a taxonomic classification on the domain level based on short sections of the amino acid sequence of an organism's proteins is possible using a transformer neural network without relying on a reference database. We have also demonstrated that it is possible to determine the frame of a short DNA sequence within an ORF using a transformer neural network without knowledge of the reference sequence or the usual comparison of a six-frame translation with a protein sequence database.

This novel application of transformer neural networks to sequence classification will support the reference-free detection of hitherto unknown bacterial and viral proteins—and, by proxy, unknown organisms—in the "unclassified" readset typically left over after metagenomic NGS data analysis. It could also support the analysis of metaproteomic experiments, allowing an initial high-level classification of peptides and thus aiding protein sequence assemblers such as SSAKE-based PASS [49] by presenting them with a reduced number of sequences with a lower likelihood of misleading overlaps.

Additionally, the ability to classify the frame of a short DNA sequence should be useful in diverse fields of study. For instance, it is very likely that this classification is possible due to an ability to recognize biologically functional amino acid sequences. In that case, the application of this classifier could allow the detection of recent frame-shift mutations in new genomes without requiring a high-quality reference sequence. It could also be integrated into gene detection algorithms to aid in the ranking of ORFs based on their likelihood of encoding a functional protein, complementing approaches such as that described by [50].

In order to allow simple integration into analysis workflows using raw NGS reads, in the future we will add an initial classification to determine which reads lie within an ORF and can thus be used for frame and taxonomic classification.

## Supporting information

**S1 Appendix. Classification with variable sequence length [51].**
(PDF)

**S2 Appendix. Frame classification with realistic NGS reads [41, 52, 53].**
(PDF)

## Acknowledgments

CK acknowledges the helpful support of Elisabeth Gasteiger concerning questions around the Swiss-Prot dataset.

We acknowledge Patrick Baumann for supporting the distributed training of the classification models.

Also available at zenodo are the trained models for frame classification [54] and taxonomic classification [55].

The code of the classification pipeline can be found in the github repository https://github.com/CBMI-HTW/TaxonomicClassification-NGS-NN.

The datasets analyzed in this study have been downloaded from the SRA and have been referenced by SRA accessions in the text.

## Author Contributions

**Conceptualization:** Benjamin Voigt, Oliver Fischer, Christian Krumnow, Christian Herta, Piotr Wojciech Dabrowski.

**Resources:** Christian Krumnow, Piotr Wojciech Dabrowski.

**Software:** Benjamin Voigt, Oliver Fischer, Christian Krumnow.

**Supervision:** Christian Herta, Piotr Wojciech Dabrowski.

**Writing – original draft:** Benjamin Voigt, Oliver Fischer, Christian Krumnow, Christian Herta, Piotr Wojciech Dabrowski.

**Writing – review & editing:** Benjamin Voigt, Oliver Fischer, Christian Krumnow, Christian Herta, Piotr Wojciech Dabrowski.

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
