## [Decision Letter · Decision Letter 0]

27 Jul 2021

PONE-D-21-19845

NGS read classification using AI

PLOS ONE

Dear Dr. Dabrowski,

Thank you for submitting your manuscript to PLOS ONE. After careful consideration, we feel that it has merit but does not fully meet PLOS ONE’s publication criteria as it currently stands. Therefore, we invite you to submit a revised version of the manuscript that addresses the points raised during the review process.

We look forward to receiving your revised manuscript.

Kind regards,

Yanbin Yin

Academic Editor

PLOS ONE

Journal Requirements:

Reviewers' comments:

Reviewer's Responses to Questions

**Comments to the Author**

1. Is the manuscript technically sound, and do the data support the conclusions?

Reviewer #1: Partly

Reviewer #2: Yes

2. Has the statistical analysis been performed appropriately and rigorously? 

Reviewer #1: No

Reviewer #2: Yes

3. Have the authors made all data underlying the findings in their manuscript fully available?

Reviewer #1: Yes

Reviewer #2: Yes

4. Is the manuscript presented in an intelligible fashion and written in standard English?

Reviewer #1: Yes

Reviewer #2: Yes

5. Review Comments to the Author

Reviewer #1: The authors present a machine learning pipeline to predict pathogens from metagenome NGS data. The pipeline consists of two parts, coding frame prediction and taxonomy classification. The machine learning models were trained and tested on test dataset and real NGS reads data. The results show the models perform not bad using ROC curve and accuracy. However, several concerns are listed as follow.

1. The authors created a dataset that only contains CDS. I think the frame can be easily predicted by translating DNA reads into proteins using six reading frames. And then check if the stop codon is included in the protein sequence. The incorrect coding frames will lead to some stop codons in the translated sequence. I strongly suggest the author compare the reading frame prediction model to other CDS prediction approaches.

2. For classification of NGS reads, I suggest the author also compare their model to other classification softwares, such as Krakens, RIEMS.

3. The purpose of the model is to predict pathogens from NGS data. The sequencing data includes sequencing error. The model should be trained and tested on NGS simulated data. E.g., the simulation data can be generated by software, such as CAMISIM.

Reviewer #2: The authors aim to predict the taxonomic classification of sequences that can not be classified by comparison to a reference database by neural network. The topic is interesting, and the method shows a good prediction performance. However, there are several problems:

1.The authors provide the pipeline in Figure 1. Although the pipeline is relatively clear, more details should be provided to better illustrate their framework and flow about neural network model.

2. Authors should pay attention to the format. For example, the model steps should be described clearly according to the format of algorithm description.

3.To study the impact of parameters, the authors should describe how to select the dimension of a single feature vector h. More comprehensive results are needed when the dimension is set to various values.

6. PLOS authors have the option to publish the peer review history of their article (what does this mean?). If published, this will include your full peer review and any attached files.

Reviewer #1: No

Reviewer #2: No

---

## [Author Response · Author response to Decision Letter 0]

13 Nov 2021

First and foremost, we would like to extend our sincere gratitude to the reviewers for their insightful and helpful comments. They have allowed us to modify and extend our work and to hopefully close the identified gaps - primarily in making the description of our approach clearer, better describing the aim of our work, and adding more realistic data and a comparison to other tools. We describe what specific action we took in response to each comment in more detail in the following sections.

## Reviever 1 ##

Reviewer's comment: The authors created a dataset that only contains CDS. I think the frame can be easily predicted by translating DNA reads into proteins using six reading frames. And then check if the stop codon is included in the protein sequence. The incorrect coding frames will lead to some stop codons in the translated sequence. I strongly suggest the author compare the reading frame prediction model to other CDS prediction approaches.

Author's response: We wholeheartedly agree with the reviewer that it is important to compare the performance of our frame prediction to existing tools in order to show whether rolling our own frame prediction makes sense. We have accordingly performed a basic benchmark against other existing tools and added a section ``Benchmarking of frame classification'' in both the ``Methods and Implementation'' (lines 241-271) and the ``Results'' (lines 324-333) sections. In short, we have found six applicable tools, out of which two were installable and could thus be used in the benchmark. On our test dataset, we outperformed both of then by a significant margin.

Reviewer's comment: For classification of NGS reads, I suggest the author also compare their model to other classification softwares, such as Krakens, RIEMS.

Author's response: While we agree with the reviewer that benchmarking is a very important factor, out proof-of-concept software is not intended as an alternative to taxonomic classification or pathogen detection tools such as Kraken or RIEMS - neither in the current state nor once it is in a production-ready state. Instead, the aim is to create a follow-up tool that will allow the analysis of the reads that still remain unclassified after the use of such tools. As such, benchmarking against these tools would not represent the intended use-case.

In order to make it clearer that we are explicitly not attempting to create yet another taxonomic profiler that solves the same problem, we have expanded the sentence in lines 70-72 to read: ``[...] after traditional metagenomic data analysis and taxonomic classification has been performed with tools such as Kraken, RIEMS, PathoScope, PAIPline or MetaMeta''.

Reviewer's comment: The purpose of the model is to predict pathogens from NGS data. The sequencing data includes sequencing error. The model should be trained and tested on NGS simulated data. E.g., the simulation data can be generated by software, such as CAMISIM.

Author's response: We thank the reviewer for pointing out this aspect and agree that investigating the stability of our scheme in more realistic settings is indeed interesting. Therefore, we added supporting information to the draft (``S2 Appendix: Frame classification with realistic NGS reads'', lines 379-410), which examines the influence of sequencing errors on the frame classification. We used ART to generate simulated NGS data. The general outcome is that sequencing errors slightly affect the accuracy of our classification, as expected. Still, the frame classification seems relatively robust, and regular error-reduction practices such as cutting off the last bases (for the sake of simplicity, we represented this by cutting off the last 50 bases of each read instead of performing a thorough evaluation of the effect of different quality trimming approaches) help to improve it.

## Reviever 2 ##

Reviewer's comment: The authors provide the pipeline in Figure 1. Although the pipeline is relatively clear, more details should be provided to better illustrate their framework and flow about neural network model.

Author's response: We agree with the reviewer that clarity is paramount in such visualizations and are very grateful for this comment. Figure 1 has been accordingly revised and now contains more details of the pipeline. We hope that it illustrates the framework and information flow more clearly now. 

Reviewer's comment: Authors should pay attention to the format. For example, the model steps should be described clearly according to the format of algorithm description.

Author's response: We thank the reviewer for pointing out that the description of the algorithm might not be sufficiently clear. Unfortunately, we have not been able to find any format specifications in the PLOS ONE Submission Guidelines regarding the illustration of algorithms. We have thus generally reworked Figure 1 to make the overall operation of the pipeline easier to understand (as per the first comment) and hope that this also satisfies the reviewer's requirements.

Reviewer's comment: To study the impact of parameters, the authors should describe how to select the dimension of a single feature vector h. More comprehensive results are needed when the dimension is set to various values.

In general, we fully agree that a more in-depth investigation of the feature vectors towards the overall performance is interesting. However, the dimension of the feature vectors results from the pre-trained language model used in the pipeline and is therefore an externally given parameter. In the case of ProtBert it is fixed to 1024. We reworked lines 156-158 to clarify this aspect. Such an evaluation would thus require retraining the whole ProtBert language model which is very - depending on the available resources even prohibitively - expensive (note that ProtBert utilized 1024 GPUs or 512 TPUs in the training process). This would also run contrary to the underlying idea of our work of investigating the power of using an existing language model, which is in part motivated by exactly this often prohibitive cost of re-training.

---

## [Decision Letter · Decision Letter 1]

6 Dec 2021

NGS read classification using AI

PONE-D-21-19845R1

Dear Dr. Dabrowski,

We’re pleased to inform you that your manuscript has been judged scientifically suitable for publication and will be formally accepted for publication once it meets all outstanding technical requirements.

Kind regards,

Yanbin Yin

Academic Editor

PLOS ONE

Additional Editor Comments (optional):

Reviewers' comments:

Reviewer's Responses to Questions

**Comments to the Author**

1. If the authors have adequately addressed your comments raised in a previous round of review and you feel that this manuscript is now acceptable for publication, you may indicate that here to bypass the “Comments to the Author” section, enter your conflict of interest statement in the “Confidential to Editor” section, and submit your "Accept" recommendation.

Reviewer #1: All comments have been addressed

Reviewer #2: All comments have been addressed

2. Is the manuscript technically sound, and do the data support the conclusions?

Reviewer #1: Yes

Reviewer #2: Yes

3. Has the statistical analysis been performed appropriately and rigorously? 

Reviewer #1: Yes

Reviewer #2: Yes

4. Have the authors made all data underlying the findings in their manuscript fully available?

Reviewer #1: Yes

Reviewer #2: Yes

5. Is the manuscript presented in an intelligible fashion and written in standard English?

Reviewer #1: Yes

Reviewer #2: Yes

6. Review Comments to the Author

Reviewer #1: (No Response)

Reviewer #2: The authors addressed reviewers’ comments well. The revised version is improved in quality. I have no further suggestions to make.

7. PLOS authors have the option to publish the peer review history of their article (what does this mean?). If published, this will include your full peer review and any attached files.

Reviewer #1: No

Reviewer #2: No

---

## [Editor Report · Acceptance letter]

10 Dec 2021

PONE-D-21-19845R1 

NGS read classification using AI 

Dear Dr. Dabrowski:

I'm pleased to inform you that your manuscript has been deemed suitable for publication in PLOS ONE. Congratulations! Your manuscript is now with our production department. 

Kind regards, 

on behalf of

Dr. Yanbin Yin 

Academic Editor

PLOS ONE